# Antigen presentation and tumor immunogenicity in cancer immunotherapy response prediction

**Shixiang Wang[1,2,3], Zaoke He[1,2,3], Xuan Wang[1,2,3], Huimin Li[1,2,3], Xue-Song Liu[1]\***

[1]School of Life Science and Technology, ShanghaiTech University, Shanghai, China; [2]Shanghai Institute of Biochemistry and Cell Biology, Chinese Academy of Sciences, Shanghai, China; [3]University of Chinese Academy of Sciences, Beijing, China

**Abstract** Immunotherapy, represented by immune checkpoint inhibitors (ICI), is transforming the treatment of cancer. However, only a small percentage of patients show response to ICI, and there is an unmet need for biomarkers that will identify patients who are more likely to respond to immunotherapy. The fundamental basis for ICI response is the immunogenicity of a tumor, which is primarily determined by tumor antigenicity and antigen presentation efficiency. Here, we propose a method to measure tumor immunogenicity score (TIGS), which combines tumor mutational burden (TMB) and an expression signature of the antigen processing and presenting machinery (APM). In both correlation with pan-cancer ICI objective response rates (ORR) and ICI clinical response prediction for individual patients, TIGS consistently showed improved performance compared to TMB and other known prediction biomarkers for ICI response. This study suggests that TIGS is an effective tumor-inherent biomarker for ICI-response prediction.

DOI: https://doi.org/10.7554/eLife.49020.001

**\*For correspondence:**
liuxs@shanghaitech.edu.cn

**Competing interests:** The authors declare that no competing interests exist.

## Introduction

Immunotherapy, represented by immune checkpoint inhibitors (ICI), including anti-PD-1 antibodies, anti-PD-L1 antibodies, anti CTLA-4 antibodies or their combinations, is transforming the treatment of cancer. Compared to conventional therapies, ICI can induce significantly improved clinical responses in patients with various types of late-stage metastatic cancers. However, the majority of unselected patients will not respond to ICI. Most tumor types show response rates below 40% to PD-1 inhibition, and the response rates of each tumor type are reported to be correlated with the tumor mutational burden (TMB) of that tumor type (*Yarchoan et al., 2017*). Multiple factors are reported to affect ICI effectiveness, including: PD-L1 expression (*Herbst et al., 2014*; *Shukuya and Carbone, 2016*), TMB (*Rizvi et al., 2015*; *Snyder et al., 2014*), DNA mismatch repair deficiency (*Le et al., 2015*), the degree of cytotoxic T cell infiltration (*Tang et al., 2016*), mutational signature (*Miao et al., 2018*; *Wang et al., 2018*), antigen presentation defects (*Chowell et al., 2018*; *Zaretsky et al., 2016*), interferon signaling (*Ayers et al., 2017*), tumor aneuploidy (*Davoli et al., 2017*) and T-cell signatures (*Jiang et al., 2018*). These biomarkers have various rates of accuracy and utility, and the identification of a robust ICI-response biomarker is still a critical challenge in the field (*Nishino et al., 2017*).

ICI help a patient's immune system to recognize and attack cancer cells. The immunogenicity of cancer cells is the fundamental determinant of ICI response. Theoretically, tumors of very low or no immunogenicity will not respond to therapeutic strategies that enhance the immune response. Hence, ICI can only be used to treat tumors that have sufficient immunogenicity. Furthermore, enhancing tumor immunogenicity can potentially transform an immunotherapy-non-responsive tumor into an immunotherapy-responsive tumor.

**eLife digest** In the last decade a new kind of cancer therapy, called immunotherapy, has changed how doctors treat cancer patients. These therapies mean that previously incurable cancers, including some skin and lung cancers, can now sometimes be cured. Immunotherapy does this by activating the patient's own immune system so that it will attack the cancer cells. But for this to work, the cancer cells, much like invading bacteria or viruses, need to be recognized as foreign.

Cancer cells contain many DNA mutations that cause the cell to make mutated proteins it would not normally make. These proteins betray the cancer cells as foreign to the immune system. The extent to which cancer cells make mutated proteins – also called the 'tumor mutational burden' – can sometimes predict whether a patient will respond to immunotherapy. In general, patients with a high mutational burden respond well to immunotherapy, but overall fewer than one in five cancer patients are cured by this treatment.

An important question is whether there are better ways of predicting if a cancer patient will respond to immunotherapy. Wang et al. have addressed this problem by adding a second variable to the prediction. Not only do cancer cells have to make mutated proteins, but these proteins also have to be 'seen' by immune cells. Cancer cells, like normal cells, have mechanisms to present protein fragments to immune cells. Wang et al. hypothesized that patients with a high mutational burden would not respond to immunotherapy if they were lacking the machinery required for presenting protein fragments.

The experiments revealed that measuring both tumor mutational burden and the levels of the machinery that presents protein fragments resulted in better predictions of patients' responses to immunotherapy than measuring tumor mutational burden alone. Additionally, this new way of predicting responses to immunotherapy was successful across many different cancer types.

The combined measurement of these two variables could be applied in clinical practice as a way to predict cancer patients' response to immunotherapy. This should allow doctors to determine which course of treatment will work best for a specific patient. The results also suggest that inducing tumor cells to produce more of the machinery that presents protein fragments to the immune system could increase their responsiveness to immunotherapy. In the future, predicting how well a patient will respond to immunotherapy could become even more accurate by incorporating additional variables.

DOI: https://doi.org/10.7554/eLife.49020.002

---

The actual immunogenicity of a tumor is not easy to measure. In theory, tumor immunogenicity is determined by the tumor cell itself, and is also influenced by factors related to the tumor microenvironment, such as the functioning of professional antigen-presenting cells like dendritic cells (DCs) (*Mellman and Steinman, 2001*). Fundamental determinants of tumor immunogenicity include tumor antigenicity, and antigen processing and presenting efficiency (*Blankenstein et al., 2012*).

Antigen presentation defects have already been shown to contribute to ICI-response failure (*Chowell et al., 2018*; *Zaretsky et al., 2016*). To measure antigen processing and presenting efficiency systematically, we applied a gene set variation analysis (GSVA) method to generate an antigen processing and presenting machinery (APM) score (APS) (*Hänzelmann et al., 2013*), which was calculated from the mRNA expression status of APM genes. Tumor immunogenicity score (TIGS) was then calculated by combining the APM score and the TMB. The antigen-presentation gene expression signature and tumor immunogenicity landscape of 32 cancer types from The Cancer Genome Atlas (TCGA) project are provided.

TIGS exhibits improved performance in both pan-cancer ICI objective response rate (ORR) correlation and accuracy of ICI clinical response prediction when compared with TMB. Our results suggest that TIGS represents a novel and effective tumor-inherent biomarker for the prediction of immunotherapy response.

# Results

## APM score definition and pan-cancer analysis

Cell surface presentation of peptides by major histocompatibility complex (MHC) class I molecules is critical to CD8$^+$ T-cell mediated adaptive immune responses, including those against tumors. The generation and loading of peptides onto MHC class I molecules require the functioning of the APM. Several steps are involved in this process, including: 1) peptide generation and trimming in the proteasome; 2) peptide transport; 3) assembly of the MHC class loading complex in the endoplasmic reticulum (ER); and 4) antigen presentation on cell surface (*Leone et al., 2013*).

The efficiency of antigen processing and presentation is one determinant of tumor immunogenicity. Here, we used the mRNA expression status of genes involved in the APM process as an indicator of the efficiency of these antigen-processing and -presenting steps. A GSVA approach was applied to measure the overall expression enrichment of APM genes (*Hänzelmann et al., 2013*). On the basis of a review paper about APM (*Leone et al., 2013*), the following genes were selected for quantification: *PSMB5, PSMB6, PSMB7, PSMB8, PSMB9, PSMB10, TAP1, TAP2, ERAP1, ERAP2, CANX, CALR, PDIA3, TAPBP, B2M, HLA-A, HLA-B* and *HLA-C* (*Figure 1—source data 1*). GSVA calculates the per sample overexpression level of a particular gene list by comparing the ranks of the genes in that list with those of all other genes. The resulting GSVA enrichment score is defined as the APS.

To explore the pan-cancer distribution pattern of APS, we analyzed about 10,000 tumors of 32 cancer types from TCGA (*Figure 1*). The boxplot in *Figure 1A* shows large variance in APS across TCGA cancer types, which uncovers significant distinction in antigen-processing and -presenting efficiency among different cancer types. This analysis is similar to a previous study of seven APM genes (*Şenbabaoğlu et al., 2016*) whose expression signature is highly correlated with the APS quantified in this study (*Figure 1—figure supplement 1*). Patient Harmonic Best Rank (PHBR) I and II scores have recently been proposed to quantify a patient's antigen presentation ability on the basis of the genotypes of their MHC class I or class II genes, respectively (*Marty Pyke et al., 2018*; *Marty et al., 2017*). However, no significant correlations can be observed between APS and PHBR scores (*Figure 1—figure supplement 1*), probably because these two methods capture different information about antigen presentation: PHBR are based on MHC genotype information, whereas APS are based on information about the expression of antigen-presentation genes. Univariate Cox regression analyses suggest that APS is associated with cancer patients' survival, and some are statistically significant (*Figure 1B*). Meta-analysis with pan-cancer hazard ratio values suggests that APS do not associate with prognosis (*Figure 1B*).

## APS determinants and associations in cancer

To identify the specific gene signatures that determine patients' APS status, we initially ran differential gene expression analysis for each TCGA cancer type on the basis of APS status. Patients with APS above the median were defined as 'APS-High', patients with APS below the median were defined as 'APS-Low'. Differential expression genes (p-value < 0.01, FDR < 0.05) were ranked by logFC from high to low and then selected for gene set enrichment analysis (GSEA) with gene sets from MSigDB (*Subramanian et al., 2005*). In results from hallmark gene sets, several gene signatures (especially interferon alpha/gamma response) were found to be enriched in most TCGA cancer types with high APS, suggesting that high APS is strongly associated with the interferon alpha/gamma signaling pathway (*Figure 2A*). GSEA using Reactome gene sets further validated this result (*Figure 2—figure supplement 1*). Interestingly, interferon gamma was reported to regulate APM gene expression (*Beatty and Paterson, 2001*; *Ikeda et al., 2002*), which is consistent with this observation.

Immune infiltration score (IIS) was calculated with GSVA using a list of marker genes for immune cell types and has been validated by the CIBERSORT method (*Şenbabaoğlu et al., 2016*) (*Figure 2—source data 1*). TIMER (*Li et al., 2016*) is another method that can accurately resolve the relative fractions of diverse cell types on the basis of gene expression profiles from complex tissues. To further validate the calculated IIS, we performed TIMER analysis (*Li et al., 2016*) and found that the TIMER results were highly correlated with the calculated IIS (*Figure 2—figure supplement 2*). Significant associations between APS and IIS at both the level of cancer types and the level of individual

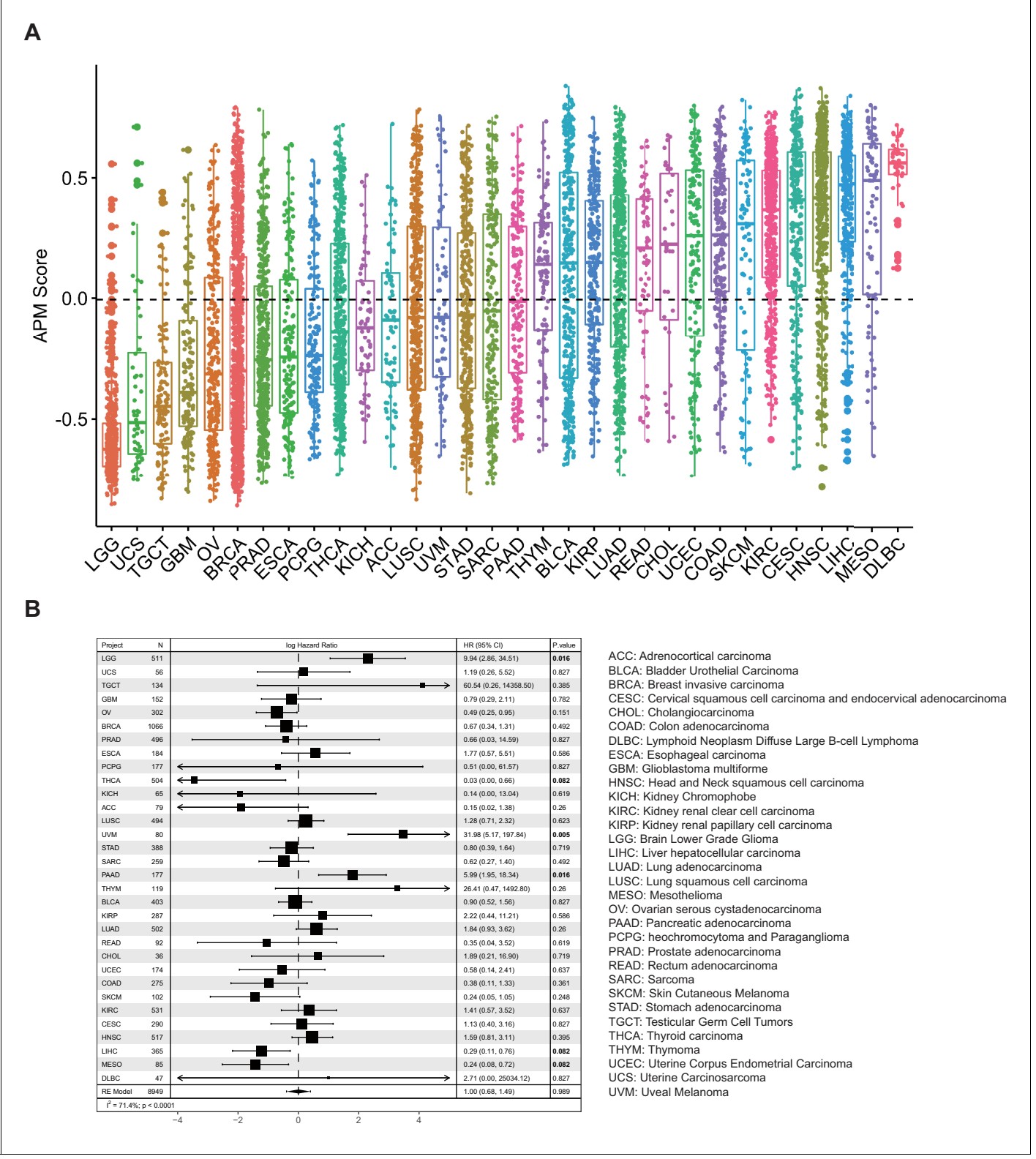

**Figure 1.** Analysis of antigen processing and presenting machinery (APM) score in 32 cancer types. (**A**) APM scores were calculated with GSVA in 32 TCGA cancer types. (**B**) Results of Cox proportional hazards regression analysis using APM score for all solid cancers. Forest plots showing $\log_e$ hazard ratio (95% confidence interval). Cox p-values are adjusted the with false discovery rate (FDR) method, p-values less than 0.1 are in bold. The pooled hazard ratio and p-value are generated by the random effect model. The statistical test for heterogeneity is also shown in the last column. Tumor types are ordered by median APM scores.

*Figure 1 continued on next page*

*Figure 1 continued*

DOI: https://doi.org/10.7554/eLife.49020.003

The following source data and figure supplement are available for figure 1:

**Source data 1.** APM gene list for GSVA.

DOI: https://doi.org/10.7554/eLife.49020.005

**Figure supplement 1.** Correlations between immune infiltration score (IIS), APS, 7-APM genes, PHBR I and PHBR II in the TCGA pan-cancer dataset.

DOI: https://doi.org/10.7554/eLife.49020.004

patients were observed (*Figure 2B and C*). The gene list for APS calculation did not overlap with the gene list for IIS calculation.

Pan-cancer distribution of TMB was also analyzed with the TCGA dataset (*Figure 2—figure supplement 3*). Different cancer types show different prognosis in relation to high TMB (*Figure 2—figure supplement 3*). Meta-analysis including all TCGA cancer types suggests that patients with high TMB tend to have poor prognosis (*Figure 2—figure supplement 3*). TMB reflects tumor antigenicity and predicted improved survival after immunotherapy. However, in cancer patients not treated with immunotherapy, high TMB tends to be associated with poor prognosis, probably because tumors accumulate mutations during progression as a result of genome instability, and consequently, high TMB is usually associated with late-stage cancer.

The immune cell subsets were assessed with both IIS and CIBERSORT (*Newman et al., 2015*) methods, and the associations between immune cell subsets with APS were analyzed further (*Figure 2—figure supplement 4*). Several types of immune cells, including cytotoxic cells, show strong correlation with APS values (*Figure 2—figure supplement 4*). TMB and IIS show relatively weak intercorrelation (*Figure 2D and E*). The significant correlation between APS and IIS could be due to the following reasons: first, the immune response coordinated by interferon signaling could regulate both APS and IIS; and second, the immunogenicity contributed by APS could stimulate immune response.

## Tumor immunogenicity score: definition and pan-cancer profiling

Tumor immunogenicity is determined by two factors: the antigenicity of tumor cells and the processing and presentation of tumor antigens. These two factors are independent, and are both required for tumor immunogenicity determination. Theoretically, tumor immunogenicity score (TIGS) can be represented as ["Tumor antigenicity"] x ["Antigen processing and presenting status"].

Non-synonymous tumor mutation and, consequently, the production of neoantigens can elicit immune response (*Schumacher and Schreiber, 2015*). Pan-cancer TMB distribution was analyzed, and log-based TMB values were found to show a Gaussian distribution (*Figure 4—figure supplement 1*). In addition, a previous study had already indicated that log(TMB) shows linear correlation with pan-cancer immunotherapy ORR (*Yarchoan et al., 2017*). Thus, we used log(TMB) as a simple representation of 'Tumor antigenicity'. APS calculated on the basis of GSVA range from −1 to 1. To multiply with tumor antigenicity, we used normalized APS values, which range from 0 to 1, as a representation of 'Antigen processing and presenting status'.

$$APS_{normalized} = \frac{APS - APS_{pancan\_min}}{APS_{pancan\_max} - APS_{pancan\_min}}$$

We calculated tumor immunogenicity score (TIGS) by using the following formula:

$$TIGS = APS_{normalized} \times log(TMB) \tag{TMB}$$

TIGS were calculated for TCGA samples for which both TMB and RNA-seq gene expression data are available (32 cancer types, 8413 samples) (*Figure 3A*). Cancer types with high TIGS include: skin cutaneous melanoma (SKCM), diffuse large B-cell lymphoma (DLBC), colon adenocarcinoma (COAD), head and neck squamous cell carcinoma (HNSC) (*Figure 3A*). Univariate Cox regression analysis suggests that TIGS is associated with cancer patients' survival, and this association is statistically significant for some cancer types (*Figure 3B*). Meta-analysis involving all TCGA cancer types suggested that high TIGS tends to be associated with a poor prognosis in patients not treated with

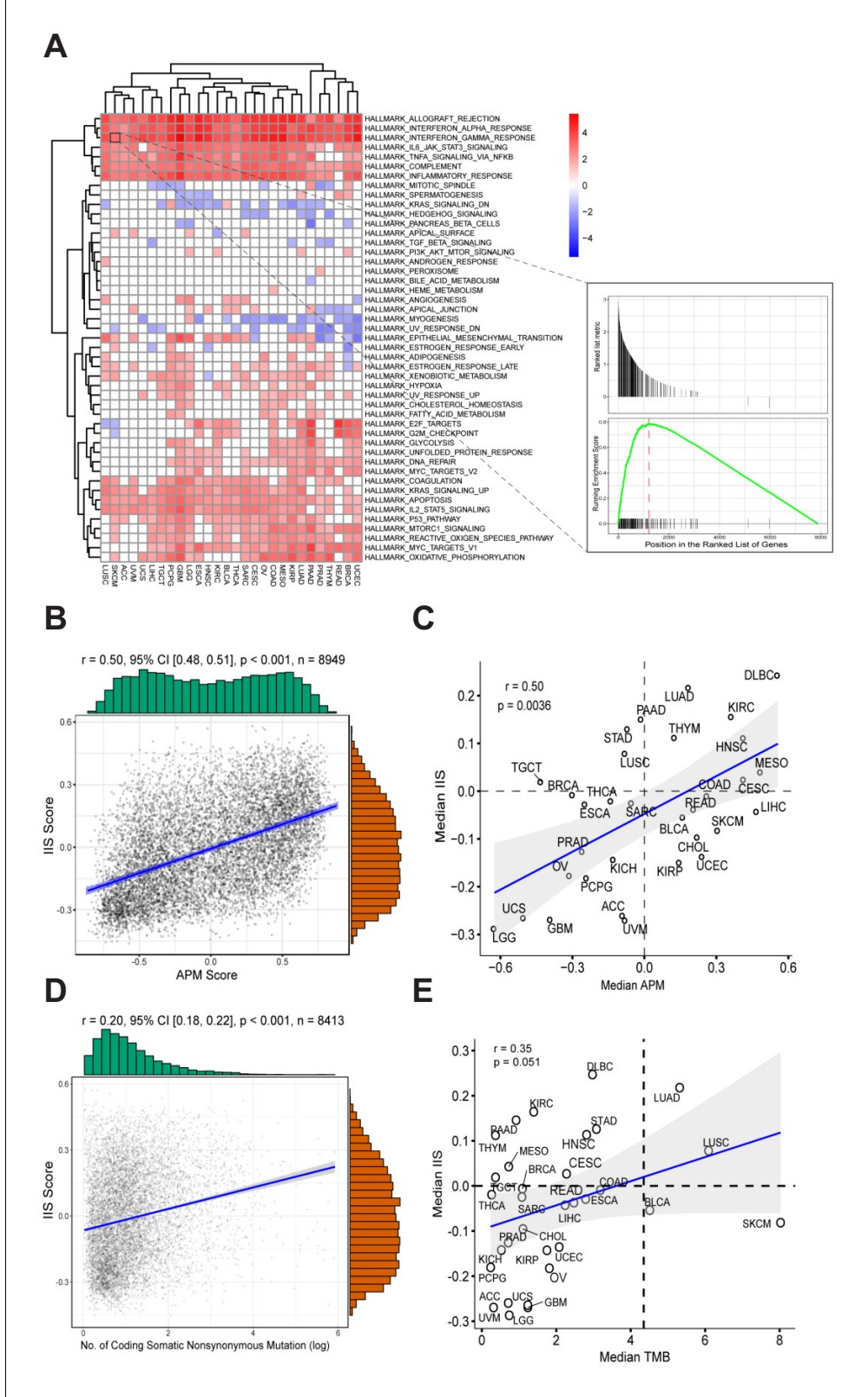

**Figure 2.** Gene expression signatures associated with high APM score. (**A**) Gene sets enriched in patients with high APM score. (**B**) Significant correlation between APM score and IIS in 8949 cancer samples. (**C**) Significant correlation between APM score and IIS in different cancer types. (**D**) Correlation between TMB and IIS in 8413 cancer samples. (**E**) Correlation between TMB and IIS in different cancer types.

*Figure 2 continued on next page*

*Figure 2 continued*

DOI: https://doi.org/10.7554/eLife.49020.006

The following source data and figure supplements are available for figure 2:

**Source data 1.** Immune cell types and corresponding signature gene lists for GSVA.
DOI: https://doi.org/10.7554/eLife.49020.011
**Figure supplement 1.** Gene sets that are enriched in 30 types of TCGA cancer patients with high APM score.
DOI: https://doi.org/10.7554/eLife.49020.007
**Figure supplement 2.** Correlation between IIS of GSVA and TIMER analysis (B_cell, etc.) results in 30 TCGA cancer types.
DOI: https://doi.org/10.7554/eLife.49020.008
**Figure supplement 3.** Analysis of tumor mutational burden (TMB) in 32 TCGA cancer types.
DOI: https://doi.org/10.7554/eLife.49020.009
**Figure supplement 4.** Immune cell subsets associated with APS were analyzed with IIS (**A**) or the CIBERSORT (**B**) method.
DOI: https://doi.org/10.7554/eLife.49020.010

immunotherapy (*Figure 3B*), which may be due to a mechanism that is the same as that which leads to high TMB.

## TIGS and pan-cancer ORR to PD-1 inhibition

Previous studies have shown that TMB can predict pan-cancer ICI ORR (*Yarchoan et al., 2017*). Here, we evaluated and compared the performance of APS, TIGS with TMB in pan-cancer ICI ORR correlation. The ORR for anti–PD-1 or anti–PD-L1 therapy were plotted against the corresponding median APS, TIGS, TMB across multiple cancer types. In an extensive literature search, we identified 25 tumor types or subtypes for which ORR data are available. For each tumor type, we pooled the response data from the largest published studies that evaluated ORR. We included only studies of anti–PD-1 or anti–PD-L1 monotherapy that enrolled at least 10 patients who were not selected for PD-L1 tumor expression. (Identified individual studies and references are available in *Figure 4—source data 1* and *Figure 4—source data 2*.)

To calculate TIGS, two different approaches can be applied. In the first approach, the APS and TMB information are obtained from different studies. This approach can include a greater number of different cancer datasets. In a second approach, all APS and TMB information is obtained from the same TCGA datasets, and in this case, fewer cancer types are available for investigation. When using the first approach, in order to calculate TIGS, the median TMB for each tumor type was obtained from a validated comprehensive genomic profiling assay that was performed and provided by Foundation Medicine (*Chalmers et al., 2017*). The APS information for 23 tumor types was calculated on the basis of TCGA datasets, whereas the APS for Merkel cell carcinoma, cutaneous squamous cell carcinoma and small-cell lung cancer were calculated on the basis of GEO microarray datasets. Significant correlations between APS, TMB, TIGS and the ORR were observed (*Figure 4*). The correlation coefficients between APS and ORR and between TMB and ORR were 0.42 (p=0.038) and 0.71 (p=6.8e-5), respectively (*Figure 4*), suggesting that 18% and 50% of the difference in the ORR across cancer types could be explained by APS and TMB, respectively. The correlation coefficient between TIGS and ORR is 0.78 (p=5.4e-6) (*Figure 4C*), indicating that 60% of the difference in ORR could be explained by TIGS. These pan-cancer ORR analyses imply that TIGS performs better than TMB or APS in correlations with immunotherapy ORR. When using the second approach for TIGS calculation, TIGS still outperformed both TMB and APS in pan-cancer ORR correlation (*Figure 4—figure supplement 1*).

## TIGS and prediction of clinical response to ICI

Compared with TMB and APS, TIGS showed improved correlation with immunotherapy ORR in various types of cancer. Here, we further evaluate the performance of TIGS in predicting ICI clinical response for individual cancer patients. Recently, several prediction biomarkers for immunotherapy response that are based on gene-expression profiling have been reported (*Ayers et al., 2017*; *Jiang et al., 2018*). *Ayers et al. (2017)* reported an IFN-γ-related mRNA expression signature that predicts clinical response to PD-1 blockade. *Benci et al.*

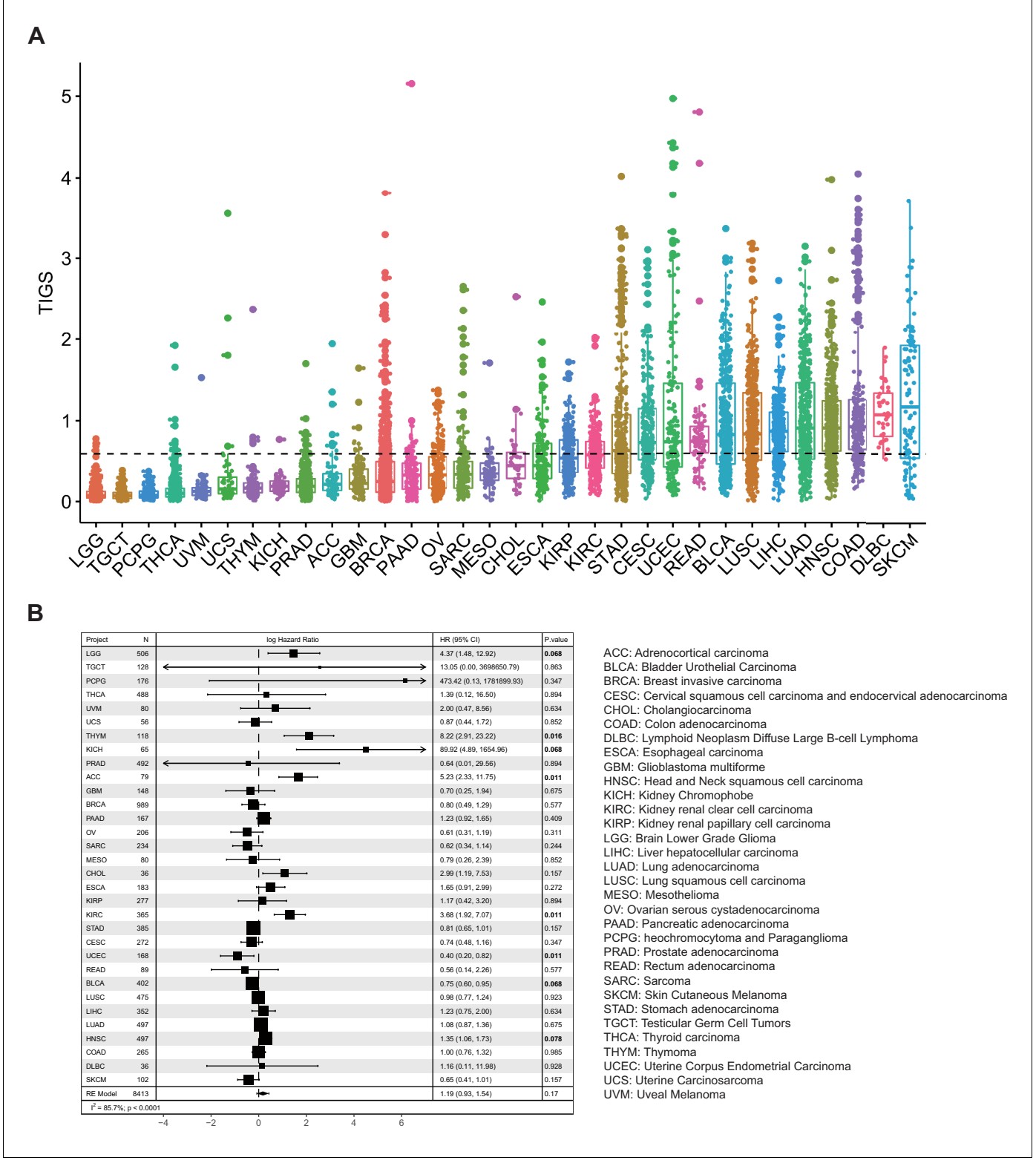

**Figure 3.** Tumor immunogenicity score (TIGS) analysis in 32 cancer types. (**A**) Analysis of TIGS in 32 cancer types. (**B**) Results of Cox proportional hazards regression analysis using TIGS for all solid cancers. Forest plots showing $\log_e$ hazard ratio (95% confidence interval). Cox p-values are adjusted with the FDR method. p-values less than 0.1 are in bold. The pooled hazard ratios and the p-values were generated using the random effect model. The statistical test for heterogeneity is also shown in the last column. Tumor types are ordered by median TIGS score.

DOI: https://doi.org/10.7554/eLife.49020.012

*Figure 3 continued on next page*

*Figure 3 continued*

The following figure supplement is available for figure 3:

**Figure supplement 1.** Pan-cancer distribution pattern of TMB in 9613 TCGA cancer samples.
DOI: https://doi.org/10.7554/eLife.49020.013

*(2019)* recently described two distinct interferon-related gene expression signatures: ISG.RS, which is associated with resistance to ICI, and by contrast, IFNG.GS, which is derived from an IFNG hallmark geneset and associated with response to ICI. *Jiang et al. (2018)* reported a T-cell dysfunction and exclusion gene expression signature (named 'TIDE' in the original paper) as a biomarker for cancer immunotherapy response. TIDE outperforms known immunotherapy biomarkers — TMB, PD-L1 expression, and interferon gamma gene expression signature — in predicting the response to immunotherapy in melanoma and lung cancer (*Jiang et al., 2018*). The predictive power of TIGS in ICI clinical response was evaluated and compared with those of TMB and biomarkers based on gene expression profiling using ICI datasets, which contain both TMB and transcriptome data for individual patients. In total, two melanoma datasets (*Hugo et al., 2016*; *Van Allen et al., 2015*) and one urothelial cancer (*Snyder et al., 2017*) dataset were available for this analysis.

To evaluate performance in predicting clinical response to ICI, we used the receiver operating characteristic (ROC) curve to measure the true-positive rates against the false-positive rates at various thresholds of TMB, TIDE or TIGS values (*Figure 5A–C*). When compared to the widely used ICI-response biomarker TMB, TIGS consistently achieved better performance in all three ICI datasets (*Figure 5A–C*). The predictive power of TIGS was comparable to that of TIDE in the two melanoma datasets. However, TIDE failed to predict response to immunotherapy in urothelial cancer, so TIGS showed better performance in the urothelial cancer dataset (*Figure 5C*). TIGS also outperforms other immunotherapy biomarkers that are based on gene expression profiling, including IIS, IFNG, ISG.RS, IFNG.GS and CD8, in all three datasets (*Figure 5D–F* and *Figure 5—figure supplement 1*). The list of genes used to calculate IFNG, ISG.RS, IFNG.GS and CD8 signatures are available *in Figure 5—source data 1*. Interestingly, APS itself also shows improved or similar prediction power when compared to other gene-expression-profiling-based biomarkers (*Figure 5D–F* and *Figure 5—figure supplement 2*). The expression profiles of randomly selected genes (named 'APSr' in *Figure 5D–F*), which were used as a negative control, failed to predict immunotherapy response in all three datasets.

In all three available datasets, Kaplan–Meier overall survival curves were further compared in patients with high vs low TIDE, TMB or TIGS level (*Figure 5G–O*). Patients with TIGS above the median were defined as 'TIGS-High' while the remaining patients were defined as 'TIGS-Low'. 'TMB-High', 'TMB-Low', 'TIGS-High' and 'TIGS-Low' were similarly defined. Comparison of survival curves showed better survival for TMB-High patients than for TMB-Low patients in all three ICI datasets, even though the difference did not reach significance in any of the three datasets, probably because of the limited sample size (*Figure 5G–I*). As defined in the original paper (*Jiang et al., 2018*), TIDE-Low indicates low tumor immune dysfunction and low immune escape, and consequently high immunotherapy response. In the *Van Allen et al. (2015)* melanoma dataset, significantly improved survival was observed in TIDE-Low patients when compared to TIDE-High patients (*Figure 5M*). In the urothelial cancer dataset (*Snyder et al., 2017*), TIDE-Low patients did not have the expected immunotherapy response (*Figure 5O*). However, TIGS-High patients showed significantly better survival curves than TIGS-Low patients in all three ICI datasets (*Figure 5J–L*). These analyses suggest that in all three available datasets, TIGS outperforms TMB and other biomarkers that are based on gene-expression profiling (TIDE, IFNG etc.) in accurately predicting clinical response to immunotherapy and in pan-cancer applicability.

## Discussion

Immunogenicity is an important inherent feature of tumor cells. This feature is determined by the tumor cell itself, and is also influenced by the tumor microenvironment. Two key determinants of tumor immunogenicity are tumor antigenicity and the ability to present such antigenicity. Here, we proposed an initial method to measure the immunogenicity of a tumor. This measured tumor immunogenicity score (TIGS) shows consistently improved correlations with immunotherapy ORR in

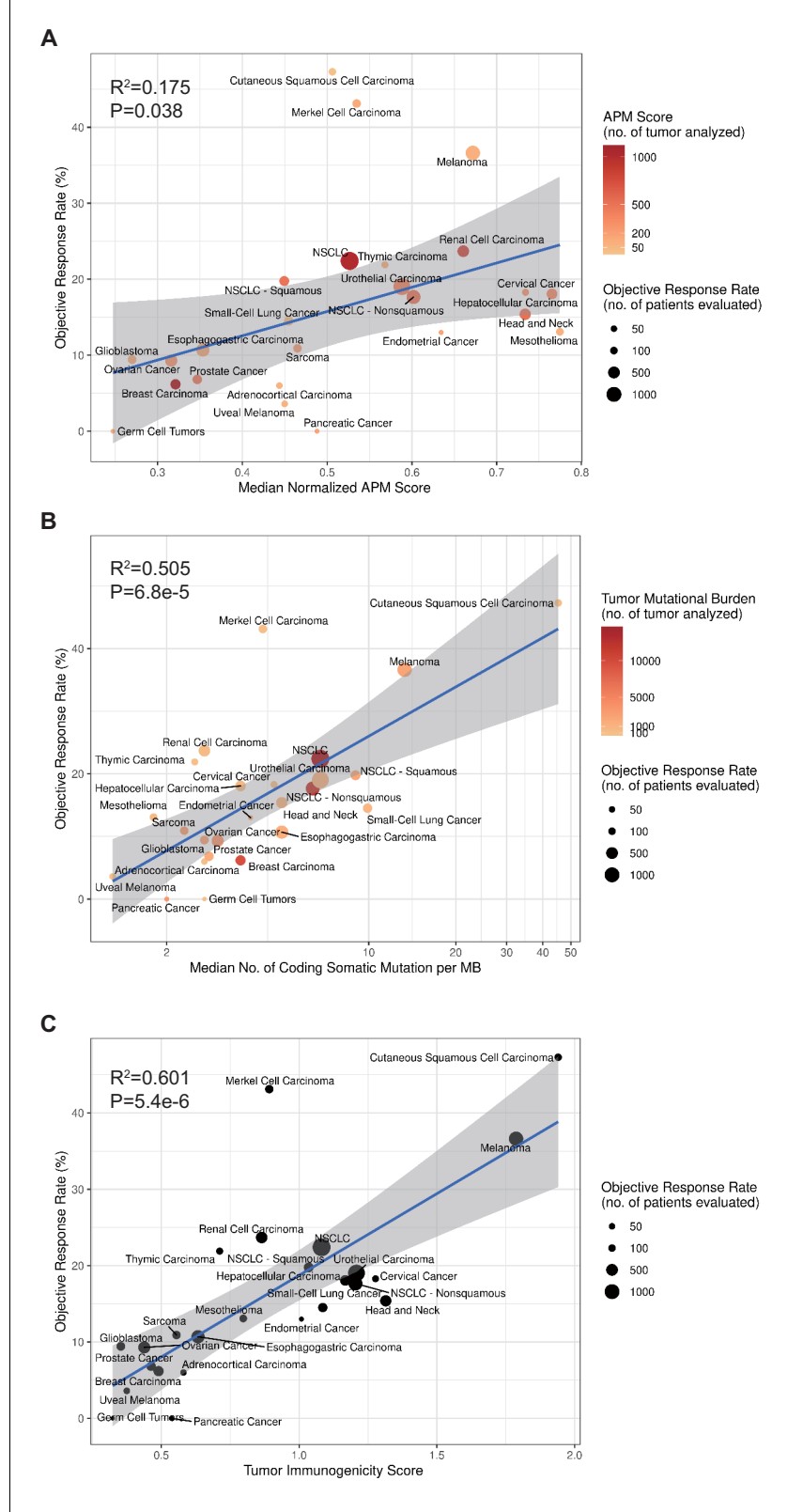

**Figure 4.** TIGS and predicted pan-cancer response rates to PD-1 inhibition. Correlation between (**A**) APS, (**B**) TMB, (**C**) TIGS and objective response rate (ORR) with anti-PD-1 or anti-PD-L1 therapy in 25 cancer types. Shown are median normalized APS (A), median number of TMB (non-synonymous mutation/MB) in log scale (B) and TIGS in 25 tumor types or subtypes among patients who received inhibitors of PD-1 or PD-L1 (C), as described in

*Figure 4 continued on next page*

*Figure 4 continued*

published studies for which data regarding the ORR are available. The number of patients who were evaluated for the ORR is shown for each tumor type (size of the circle), along with the number of tumor samples that were analyzed to calculate the APS, TMB or TIGS (degree of shading of the circle).

DOI: https://doi.org/10.7554/eLife.49020.014

The following source data and figure supplement are available for figure 4:

**Source data 1.** List of citations for individual studies used in pooled analysis of objective response rate.
DOI: https://doi.org/10.7554/eLife.49020.016
**Source data 2.** Summary of pooled ORR, median TMB and median APS by tumor type or subtype.
DOI: https://doi.org/10.7554/eLife.49020.017
**Figure supplement 1.** TIGS and predicted pan-cancer response rates to PD-1 inhibition.
DOI: https://doi.org/10.7554/eLife.49020.015

various types of cancer when compared to TMB. TIGS also shows improved performance in ICI clinical response prediction when compared with TMB and other biomarkers that are based on gene expression profiling (TIDE, interferon gamma signature and so on) in both prediction accuracy and pan-cancer applicability. Furthermore, our tumor-immunogenicity-based biomarker could guide the treatment to transform some ICI-non-responsive tumors into ICI-responsive tumors. Stimulating the APM pathway could enhance tumor immunogenicity, and possibly ICI responsiveness.

Our study demonstrates that TIGS is an effective biomarker for ICI-response prediction. TIGS capture two key aspects of tumor immunogenicity, antigen presentation and tumor antigenicity, which could be the reason for its improved performance in ICI-response prediction when compared to known biomarkers. Furthermore, our formula for TIGS calculation can point to a new way to transform some ICI-non-responsive tumors into responsive tumors by enhancing the tumor immunogenicity. One approach is to enhance the efficiency of antigen presentation. Our GSEA indicates that interferon signaling is the top gene signature associated with APS-High, and interferon signaling has been reported to influence APM gene expression (*Beatty and Paterson, 2001*; *Ikeda et al., 2002*). We may enhance antigen presentation by stimulating interferon signaling in patients who are initially not responsive to ICI, especially in cancer types that have low APS, such as prostate cancer and breast cancer.

Our study identified several cancer types in which antigen presentation status makes a significant contribution in ICI response. Breast cancer and prostate cancer have usual TMB but fairly low ICI-response rates, probably because of low APS; renal clear cell carcinoma has good ICI response rate, possibly as a result of high APS. Furthermore, our linear correlation formula — $ORR = 21.4 \times TIGS - 2.7$ (this formula is based on the data in *Figure 4C*) — can be used to make hypotheses with respect to the ORR in tumor types for which anti–PD-1 therapy has not been explored. For example, we anticipate a clinically meaningful ORR of 12.3% (95% confidence interval [CI], 8.8% to 15.8%) for uterine corpus endometrial carcinoma (UCEC) on the basis of a median TIGS of 0.7.

This study reports the first quantification of tumor immunogenicity. Several situations need to be considered for future improvement of this quantification. First, other factors including tumor germline antigen, copy number variation status, tumor purity and intra-tumor heterogeneity should also be considered to enable more accurate measurement of the antigenicity of tumor cells. Second, for quantifying antigen presentation efficiency, APM protein expression and function assessment will be more accurate than APM mRNA expression measurement. Third, other factors that influence TIGS should also be considered, including the function of professional antigen presentation cells (dendritic cells for example) in the immune microenvironment.

This manuscript primarily focused on the cytosolic or endogenous neoantigen presentation pathway mediated by MHC class I. This does not mean that the potential neoantigen presentation by MHC class II is not important, and further studies are needed to improve the methods for the quantification of antigen presentation in cancer patients. In addition, a sex difference in the predictive power of TMB has been reported recently in lung cancer (*Wang et al., 2019b*; *Wang et al., 2019c*). To explore the potential sex difference in TIGS's predictive power, we need larger datasets with more patients.

TIGS is an extension and enhancement of the immunotherapy biomarker TMB. TIGS is tumor cell-based, and is distinct from the recent immunotherapy biomarkers immunophenoscore

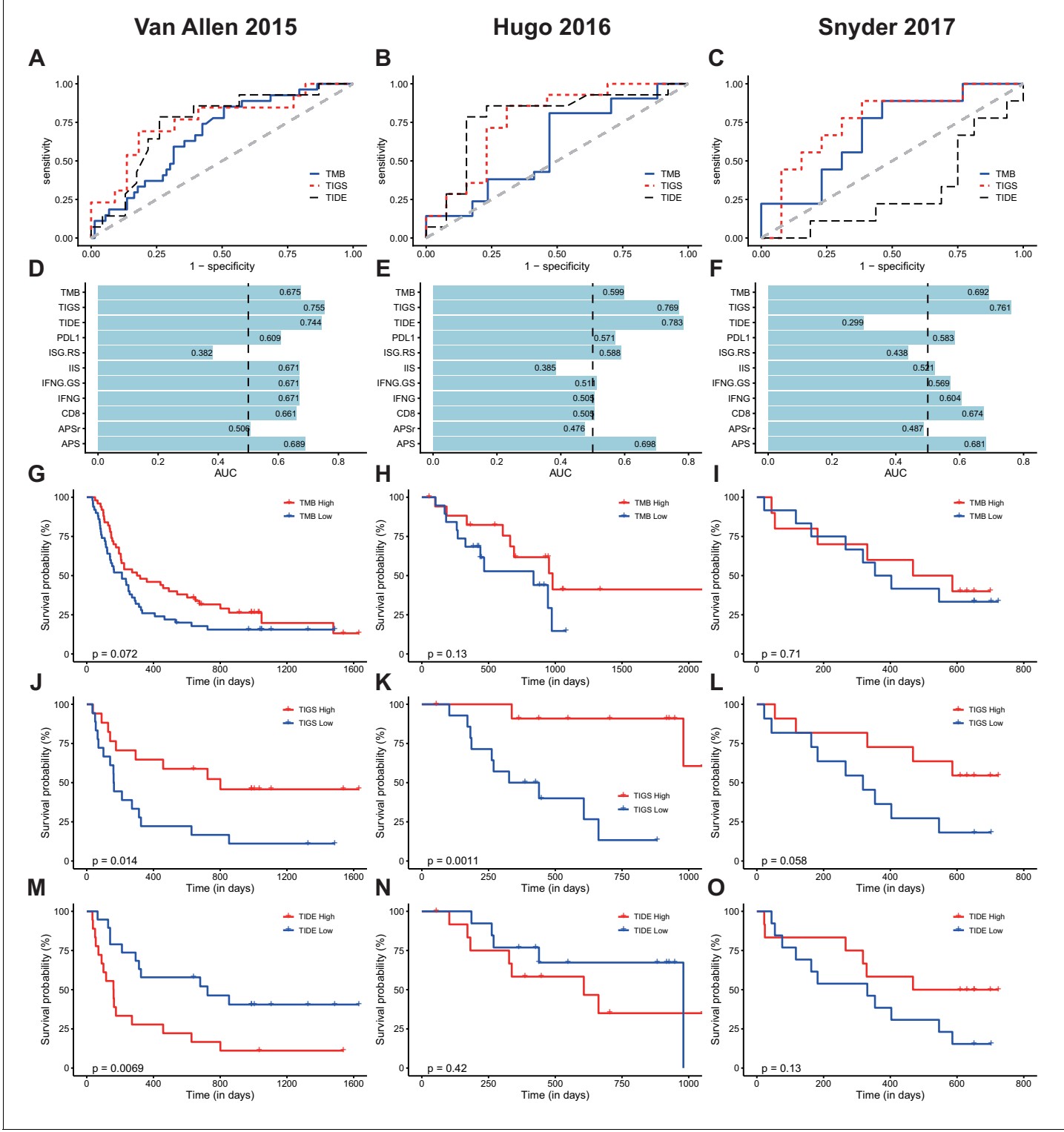

**Figure 5.** TIGS predicts clinical response to ICI immunotherapy. (**A**) ROC curves for the performance of TMB, TIDE and TIGS in predicting anti-CTLA4 Immunotherapy response in 35 melanoma patients (dataset from *Van Allen et al., 2015*). (**B**) ROC curves for the performance of TMB, TIDE and TIGS in predicting anti-PD-1 immunotherapy response in 27 melanoma patients (dataset from *Hugo et al., 2016*). (**C**) ROC curves for the performance of TMB, TIGS and TIDE in predicting anti-PD-L1 immunotherapy response in 22 urothelial cancer patients (dataset from *Snyder et al., 2017*). (**D–F**) AUC values of TMB, TIGS, TIDE, PDL1, immune infiltration score (IIS), interferon gamma gene expression signature (IFNG), CD8, APS and random genes as negative control for APS quantification (APSr) in the *Van Allen et al. (2015)* dataset (D), the *Hugo et al. (2016)* dataset (E) and the *Snyder et al.*

*Figure 5 continued on next page*

*Figure 5 continued*
*(2017)* dataset (F). The performance of a random predictor (AUC = 0.5) is represented by the dashed line. (G,J,M) Patients were grouped on the basis of TMB (G), TIGS (J) or TIDE (M) status. The Kaplan–Meier (KM) overall survival curves were compared between TMB-High and TMB-Low (100 patients), between TIGS-High vs TIGS-Low (35 patients) or between TIDE-High and TIDE-Low (37 patients) in the *Van Allen et al. (2015)* dataset. (H,K,N) Patients were grouped on the basis of TMB (H), TIGS (K) or TIDE (N) status. The KM overall survival curves were compared between TMB-High and TMB-Low (37 patients), between TIGS-High and TIGS-Low (26 patients) or between TIDE-High and TIDE-Low (26 patients) in the *Hugo et al. (2016)* dataset. (I,L,O) Patients were grouped on the basis of TMB (I), TIGS (L) or TIDE (O) status. The KM overall survival curves were compared between TMB-High and TMB-Low (22 patients), TIGS-High and TIGS-Low (22 patients) or TIDE-High and TIDE-Low (25 patients) in the *Snyder et al. (2017)* dataset.
DOI: https://doi.org/10.7554/eLife.49020.018
The following source data and figure supplements are available for figure 5:
**Source data 1.** List of genes in the lists used for CD8, IFNG, ISG.RS and IFNG.GS signature calculation.
DOI: https://doi.org/10.7554/eLife.49020.021
**Figure supplement 1.** ROC curves for the performance of APS, CD8, IFNG, IIS, PDL1 and TIGS in predicting immunotherapy response in the *Van Allen et al. (2015)* melanoma dataset (A), the *Hugo et al. (2016)* melanoma dataset (B) and the *Snyder et al. (2017)* urothelial cancer dataset (C).
DOI: https://doi.org/10.7554/eLife.49020.019
**Figure supplement 2.** APS in predicting the clinical response to immunotherapy.
DOI: https://doi.org/10.7554/eLife.49020.020

(*Charoentong et al., 2017*) or T-cell dysfunction and exclusion signature (*Jiang et al., 2018*). Both of these ICI biomarkers are based on tumor immune microenvironment. As a tumor inherent biomarker, TIGS can not only be used for predicting immunotherapy response, but also point ways to manipulate the immunogenicity of tumors, and consequently the response to immunotherapy.

## Materials and methods

### Pan-cancer clinical, gene expression and mutation data

The pancan normalized gene-level RNA-Seq data and clinical information for 33 TCGA cohorts were downloaded from UCSC Xena (https://xenabrowser.net/) with R package UCSCXenaTools (*Wang and Liu, 2019a*). Samples with 'pathologic stage' 0 or X were filtered out and only 'sample type' is 'Primary Tumor' (32 cancer types, N = 9109) were saved for further analysis. Pre-compiled, curated somatic mutations (MC3 version) for TCGA cohorts were downloaded by the R package TCGAmutations (*Ellrott et al., 2018*). Microarray gene expression datasets for Merkel cell carcinoma, cutaneous squamous carcinoma and small cell lung cancer were downloaded from the GEO database via R package GEOquery (*Davis and Meltzer, 2007*). Specifically, GSE39612 (*Harms et al., 2013*), GSE22396 (*Paulson et al., 2011*), GSE36150 (*Masterson et al., 2014*), GSE50451 (*Daily et al., 2015*), GSE99316 (*Sato et al., 2013*) were identified and downloaded.

### Implementation of GSVA

APM gene expression status and infiltration levels for immune cell types were quantified using the GSVA method implemented in the R package GSVA (*Hänzelmann et al., 2013*). RNA-Seq or microarray datasets were provided as input and output is a near-Gaussian list of decimals that can be used in visualization or downstream statistical analysis. Lists of genes for quantifying immune cell types were as previously described (*Şenbabaoğlu et al., 2016*). Gene lists for APM score and quantification of immune cell type are provided in *Figure 1—source data 1* and *Figure 2—source data 1*.

### Calculation of immune infiltration score

The immune infiltration score (IIS) for a sample was defined as the mean of standardized values for macrophages, DC subsets (total, plasmacytoid, immature, activated), B cells, cytotoxic cells, eosinophils, mast cells, neutrophils, NK cell subsets (total, CD56 bright, CD56 dim), and all T-cell subsets (CD8 T, T helper, T central and effector memory, Th1, Th2, Th17, and Treg cells). In vitro validation with multiplex immunofluorescence, in silico validation using simulated mixing proportions and comparison between CIBERSORT (*Newman et al., 2015*) and IIS have been described previously (*Şenbabaoğlu et al., 2016*). TIMER (*Li et al., 2016*) is another method that can accurately resolve

the relative fractions of diverse cell types on the basis of gene expression profiles from complex tissues. To further validate the calculated IIS, we performed TIMER analysis and found that the result of TIMER was highly correlated with the calculated IIS (*Figure 2—figure supplement 1*).

## APM score normalization for TIGS calculation

Original APM scores (APS) from GSVA are in the range of $-1$ to 1. To calculate TIGS, original APM score from GSVA implementation was rescaled by the minimal and maximal APM score from TCGA Pan-cancer analysis. The formula is

$$APS_{normalized} = \frac{APS - APS_{pancan\_min}}{APS_{pancan\_max} - APS_{pancan\_min}}$$

where $APS_{pancan\_min}$ is the minimal APM score among TCGA pan-cancer samples; and $APS_{pancan\_max}$ is the maximal APM score among TCGA pan-cancer samples. The normalized APM scores are in the range of 0 to 1. The normalized APS is set to 0 if a loss of function mutation exists in the *B2M* gene.

## Normalization of TMB data for TIGS calculation

TMB was defined as the number of non-synonymous alterations per megabase (Mb) of genome examined. As reported previously (*Chalmers et al., 2017*), we used 38 Mb as the estimate of the exome size. For studies reporting mutation number from whole exome sequencing, the normalized TMB = (whole exome non-synonymous mutations)/(38 Mb).

## TIGS calculation

We calculated TIGS as following:

$$\mathrm{TIGS} = \mathrm{APS}_{normalized} \times \log(\mathrm{TMB})$$

The natural logarithm was used here. Notably, some tumors have a TMB level below one mutation/Mb, so to avoid a negative number in quantifying 'tumor antigenicity', we added a pseudo count of one to normalized TMB. So the TIGS formula is:

$$TIGS = APS_{normalized} \times \ln(TMB + 1)$$

or

$$TIGS = APS_{normalized} \times \ln\left(\frac{\text{whole exome mutation number}}{38} + 1\right)$$

## Immunotherapy clinical studies search strategy

The dataset search strategy for assessment of cancer immunotherapy ORR) assessment has been described previously (*Yarchoan et al., 2017*). We searched MEDLINE (from January 1, 2012 to September 1, 2018), as well as abstracts in the American Society of Clinical Oncology (ASCO), the European Society for Medical Oncology (ESMO), and the American Association for Cancer Research (AACR), to identify clinical studies for anti-PD1 or anti-PDL1 therapy in various tumor types or subtypes. We searched for clinical trials using the following keywords: nivolumab, BMS-936558, pembrolizumab, MK-3475, atezolizumab, MPDL3280A, durvalumab, MEDI4736, avelumab, MSB0010718C, BMS-936559, cemiplimab, and REGN2810. We excluded studies that enrolled fewer than 10 participants, studies that investigated anti-PD-(L)one therapies only in combination with other agents, and studies that selected patients on the basis of PD-L1 expression or other immune-related biomarkers. Of the remaining studies, only the largest published study for each anti-PD-(L) one therapy was included in the final assessment of pooled ORR for each tumor type or subtype. The final identified individual studies are summarized and presented in *Figure 4—source data 1*. The TMB information for major solid tumor types or subtypes has been described previously (*Chalmers et al., 2017*). The APS of most tumor types or subtypes are based on TCGA RNA-seq data, except those for Merkel cell carcinoma, cutaneous squamous carcinoma and small cell lung cancer, which do not have available TCGA RNA-seq data. For these cancer types, the GEO datasets GSE39612, GSE22396, GSE36150, GSE50451, GSE99316 were used to generate APS. In total, 28 cancer types have both TMB and ORR values, and 25 of them also have transcriptome data that can

be used for calculating APS. Therefore, TIGS were calculated for these 25 cancer types which have both TMB and APS information available (*Figure 4—source data 2*). Linear regression models were constructed to correlate ORR with APS, TMB and TIGS for each of the cancer types or subtypes.

## Collection and analysis of immunotherapy genomics datasets

To evaluate the power of TIGS to predict clinical response to ICIs, we searched PubMed for ICI clinical studies for which TMB and gene transcriptome information was available for individual patients. In total, three datasets were identified after this search. The *Van Allen et al. (2015)* dataset was downloaded from the supplementary files of reference (*Van Allen et al., 2015*). This dataset related to CTLA-4 blockade in metastatic melanoma, and defined 'clinical benefit' using a composite end point of complete response or partial response to CTLA-4 blockade as assessed by RECIST criteria or stable disease by RECIST criteria with overall survival greater than 1 year, 'no clinical benefit' was defined as progressive disease by RECIST criteria or stable disease with overall survival less than 1 year (*Van Allen et al., 2015*). The *Hugo et al. (2016)* dataset was downloaded from the supplementary files of reference (*Hugo et al., 2016*). This dataset related to anti-PD-1 therapy in metastatic melanoma: responding tumors were derived from patients who have complete or partial responses or stable disease in response to anti-PD-1 therapy; non-responding tumors were derived from patients who had progressive disease (*Hugo et al., 2016*). The *Snyder et al. (2017)* dataset (*Snyder et al., 2017*) was downloaded from https://github.com/hammerlab/multi-omic-urothelial-anti-pdl1. This dataset related to PD-L1 blockade in urothelial cancer: durable clinical benefit was defined as progression-free survival >6 months (*Snyder et al., 2017*). RNA-Seq data were used to calculate the APS for each patient. Only patients for whom both APS and TMB value were available were used to calculate the TIGS. The median of TMB or TIGS was used as the threshold to separate the TMB-High and TMB-Low groups or the TIGS-High and TIGS-Low group in Kaplan-Meier overall survival curve analysis.

## Performance comparison on predicting immunotherapy response

The immunotherapy clinical response prediction performance of TIGS and APS have been compared with those of the following biomarkers: TMB, TIDE, IFNG, IFNG.GS, ISG.RS, PDL1, IIS, and CD8. The TIDE score was calculated using online software that is available on the website http://tide.dfci.harvard.edu. We followed the instructions on the website to generate input data for TIDE score calculation and exported the results to CSV files. The TIDE scores in the result files were used to predict response. The calculation of scores for the gene-expression-profiling-based biomarkers (i.e. IFNG, CD8, and PDL1) has been described by *Jiang et al. (2018)*. The average expression values among all members defined by the original publications were used to quantify each biomarker. The interferon gamma gene expression signature (*Ayers et al., 2017*) (IFNG) used genes *IFNG*, *STAT1*, *IDO1*, *CXCL10*, *CXCL9*, and *HLA-DRA*. The calculation of IFNG.GS and ISG.RS scores were previously described in *Benci et al. (2019)*. CD8 used genes *CD8A* and *CD8B*. PDL1 used gene *CD274*. As a negative control, we performed GSVA with 18 randomly selected genes, and the resulting score was named 'APSr' here. This GSVA with random genes was repeated for 100 times, and APSr were used to predict immunotherapy response. The average AUC of these 100 APSr is shown.

## Statistical analysis

Univariate cox analysis was performed by R package survival. P values were adjusted using the FDR method, and FDR < 0.1 is considered statistically significant. Hazard ratios and their 95% confidence intervals for TCGA cancer types were collected and used for meta-analysis with the random effect model in the R package metafor (*Viechtbauer, 2010*). The receiver operator characteristic (ROC) curve was generated by plotting the rate of response at various threshold settings of TMB, TIDE or TIGS within the R package pROC (*Robin et al., 2011*). The area under the curve (AUC) was reported for each analysis. On the basis of the median of TMB, TIDE or TIGS, we separated patients into High and Low group in the survival analysis. Keplan-Meier curves of overall survival were thus plotted with log-rank test p-value in the R package ggpubr. For GSEA enrichment analysis, we compared samples that had APS above the median with those that had APS below the median across TCGA tumor types using the limma package (*Ritchie et al., 2015*). Genes with p-value < 0.01 and FDR < 0.05

were ranked by logFC from top to bottom and then inputted into the GSEA function of the R package clusterProfiler (*Yu et al., 2012*) with custom gene sets downloaded from Molecular Signature Database v6.2 (*Liberzon et al., 2015*; *Subramanian et al., 2005*). Normalized enrichment score (NES) was used to rank the differentially enriched gene sets. Correlation analysis was performed using the spearman method. All reported p-values are two-tailed, and for all analyses, p<=0.05 is considered statistically significant, unless otherwise specified. Statistical analyses were performed using R (version 3.6.0).

## Data availability

All of the code and data used to generate the figures are freely available at https://github.com/XSLiuLab/tumor-immunogenicity-score (*Wang, 2019*; copy archived at https://github.com/elifesciences-publications/tumor-immunogenicity-score). Analyses can be read online at https://xsliulab.github.io/tumor-immunogenicity-score/. Source data files have been provided for *Figures 1*, *2*, *4* and *5*.

## Acknowledgements

We thank the authors and participating patients of the immunotherapy publications for providing the data used for this analysis. Our gratitude is also extended to the TCGA project for making cancer genomics data available for analysis. We thank Raymond Shuter for editing the text. Thanks also to the ShanghaiTech University High Performance Computing Public Service Platform for providing computing services. Thanks also to other members of Liu lab for helpful discussions.

## Additional information

### Funding

| Funder | Grant reference number | Author |
| --- | --- | --- |
| National Natural Science Foundation of China | 31771373 | Xue-Song Liu |

The funders had no role in study design, data collection and interpretation, or the decision to submit the work for publication.

### Author contributions

Shixiang Wang, Resources, Data curation, Software, Formal analysis, Validation, Investigation, Visualization, Methodology; Zaoke He, Xuan Wang, Huimin Li, Investigation, Methodology; Xue-Song Liu, Conceptualization, Resources, Formal analysis, Supervision, Funding acquisition, Validation, Investigation, Methodology, Writing—original draft, Project administration, Writing—review and editing

### Author ORCIDs

Shixiang Wang (iD) https://orcid.org/0000-0001-9855-7357
Xue-Song Liu (iD) https://orcid.org/0000-0002-7736-0077

### Decision letter and Author response

Decision letter https://doi.org/10.7554/eLife.49020.035
Author response https://doi.org/10.7554/eLife.49020.036

## Additional files

### Supplementary files

• Transparent reporting form DOI: https://doi.org/10.7554/eLife.49020.022

## Data availability

All the code and data used to generate the figures are freely available at https://github.com/XSLiu-Lab/tumor-immunogenicity-score (copy archived at https://github.com/elifesciences-publications/tumor-immunogenicity-score). Analyses can be read online at https://xsliulab.github.io/tumor-immu-nogenicity-score/. Source data files have been provided for Figures 1, 2, 4 and 5.

The following previously published datasets were used:

| Author(s) | Year | Dataset title | Dataset URL | Database and Identifier |
|---|---|---|---|---|
| Harms P, Bichak-jian C | 2013 | Distinct gene expression profiles of viral- and non-viral associated Merkel cell carcinoma revealed by transcriptome analysis | https://www.ncbi.nlm.nih.gov/geo/query/acc.cgi?acc=GSE39612 | NCBI Gene Expression Omnibus, GSE39612 |
| Paulson KG, Iyer JG, Schelter J, Cleary MA, Hard-wick J, Nghiem P | 2011 | Gene expression analysis of Merkel Cell Carcinoma | https://www.ncbi.nlm.nih.gov/geo/query/acc.cgi?acc=GSE22396 | NCBI Gene Expression Omnibus, GSE22396 |
| Masterson L, Thi-bodeau BJ, Fortier LE, Geddes TJ, Pruetz BL, Keidan R, Wilson GD | 2014 | Gene expression changes associated with prognosis of Merkel cell carcinoma | https://www.ncbi.nlm.nih.gov/geo/query/acc.cgi?acc=GSE36150 | NCBI Gene Expression Omnibus, GSE36150 |
| Brownell I, Daily K | 2015 | Microarray analysis of Merkel cell carcinoma (MCC) tumors, small cell lung cancer (SCLC) tumors, and MCC cell lines | https://www.ncbi.nlm.nih.gov/geo/query/acc.cgi?acc=GSE50451 | NCBI Gene Expression Omnibus, GSE50451 |
| Sato T, Kaneda A, Tsuji S, Isagawa T, Yamamoto S, Fujita T, Yamanaka R, Tanaka Y, Nukiwa T, Marquez VE, Ishikawa Y, Ichinose M, Aburatani H | 2013 | Gene repression and ChIP-seq in Human Small Cell Lung Cancer | https://www.ncbi.nlm.nih.gov/geo/query/acc.cgi?acc=GSE99316 | NCBI Gene Expression Omnibus, GSE99316 |

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
