## [Decision Letter]

**Acceptance summary:**

Identification of biomarkers that can predict which patients will benefit from immune checkpoint inhibition therapy is clinically important. Wang et al., describe a new computational method to identify responders to immune checkpoint inhibitors by calculating a tumor immunogenicity score (TIGS). TIGS combines tumor mutational burden (TMB) with a gene set score of 18 genes associated with MHC class I Antigen Presentation Machinery (APM) score. They describe the APM score across cancer types in TCGA and correlate APM with other gene expression pathways and immune cell infiltration across cancers. In both a pan-cancer analysis of ICI objective response rates and an ICI clinical response prediction for individual patients, they show that TIGS predicts response to ICI better than TMB alone, PD-L1, immune infiltrate, or Interferon-gene signatures, and somewhat better than the TIDE method based on T cell dysfunction and exclusion gene expression signature. TIGS is a tumor inherent biomarker and may be valuable in predicting response to immunotherapy as well as guiding ways to enhance the immunogenicity of tumors.

**Decision letter after peer review:**

Thank you for submitting your article "Antigen presentation and tumor immunogenicity in cancer immunotherapy response prediction" for consideration by *eLife*. Your article has been reviewed by three peer reviewers, and the evaluation has been overseen by a Reviewing Editor and Tadatsugu Taniguchi as the Senior Editor. The following individual involved in review of your submission has agreed to reveal their identity: Hongbin Ji (Reviewer #1).

The reviewers have discussed the reviews with one another and the Reviewing Editor has drafted this decision to help you prepare a revised submission.

Summary:

Wang et al. have developed a method named tumor immunogenicity score (TIGS) that combines tumor mutational burden (TMB) and antigen processing and presenting machinery gene expression signature to measure tumor immunogenicity. They found that TIGS could outperform TMB and other known ICI response prediction biomarkers in both pan-cancer ICI objective response rates correlation and ICI clinical response prediction.

Essential revisions:

1) The results of Şenbabaoğlu should be cited more fully and their 18 gene APS compared to the 7 gene Şenbabaoğlu APM geneset. The antigen presenting set is all MHC class I. What is the reasoning behind not including MHC class II genes like DRB1, DRB2, CIITA?

2) TIGS should be compared to TIDE + TMB. Inflamed gene expression signatures have not been shown to be a highly predictive biomarker so I think that comparisons to interferon signatures will show superiority but not guide any advance in our thinking.

3) Multiple reviewers were perplexed by why high TIGs is associated with good outcome in some tumor types but poorer outcomes in other tumor types (Figure 3B). Expand the discussion here.

4) Improve the Cox regression analysis as suggested and make the definition of high or low marker expression consistent throughout.

5) The suggestion to apply the new analyses to many cancer types for ICI response prediction is limited by the available datasets that have all of the required information. The 3 datasets analyzed are what is available and these have been done.

6) Indicate the source or reference for their linear correlation formula – "objective response rate = 21.4 ×TIGS – 2.7, ".

7) If a patient has a beta2m mutation, is this captured by the APM signature and TIGS method?

Reviewer #1:

Identification of the biomarkers that predict which patients may benefit from immune checkpoint inhibition therapy is clinically important. In this study, Wang et al. developed a method named tumor immunogenicity score (TIGS) that combined tumor mutational burden (TMB) and antigen processing and presenting machinery gene expression signature to measure tumor immunogenicity. They found that TIGS could outperform TMB and other known ICI response prediction biomarkers in both pan-cancer ICI objective response rates correlation and ICI clinical response prediction. Thus, they proposed that TIGS is a potential tumor inherent biomarker for ICI response prediction. Overall, the study is novel and interesting. I list some of my concerns below.

1) The definition of high or low marker expression should be consistent throughout. For example, the authors defined patients with APS of first quartile as "APS-High", and those at the fourth quartile as "APS-Low". In contrast, they defined patients with TIGS above the median as "TIGS-high", and the remaining as "TIGS-low". Similar issues exist for the definition of TMB. Please make the correction and explain the rationale behind.

2) In Figure 3B, Cox regression analysis show that high TIGS is significantly associated with poor survival of patients in several types of malignancies such as adrenocortical carcinoma (HR=5.23, p=0.00105), Kidney Chromophobe (HR=89.9, p=0.01408), Thymoma (HR=8.22, p=0.00198)…etc. How to explain this phenomena given that TIGS reflects tumor immunogenicity and high TIGS predicts favorable prognosis in patients following immunotherapy.

3) In this study, the authors demonstrate an improved predictive power of TIGS in ICI clinical response when compared to TMB and other gene expression profiling based biomarkers, such as TIDE, IIS, IFNΓ. The authors should discuss the potential mechanisms underlying the superior performance of TIGS for immunotherapy clinical response prediction. The authors state "Furthermore, our linear correlation formula – objective response rate = 21.4 ×TIGS – 2.7, – can be used to.…". Please also indicate the source or reference for this formula.

Reviewer #2:

This well-written article describes a score, the Tumor Immunogenicity Score (TIGS) which combines tumor mutation burden (TMB) and a gene set score of 18 genes associated with Antigen Presentation Machinery (APM) score (APS). The 18 genes were described by Leone et al., 2013) and are PSMB5, PSMB6, PSMB7, PSMB8, PSMB9, PSMB10, TAP1, TAP2, ERAP1, ERAP2, CANX, CALR, PDIA3, TAPBP, B2M, HLA-A, HLA-B, and HLA-C. The author show that the TIGS (product of the natural log of TMB and the normalized APS) marginally outperforms TMB or APS in A) linear associated with Objective Response Rate (ORR), which is a rank order of tissue response to PD1 and PDL1 therapy and B) prediction of response in 2 tissue (Melanoma, Urothelial) in 3 studies (Van Allen, 2015, Hugo et al., 2016 and Snyder et al., 2017).

This article represents an extension to Şenbabaoğlu et al., 2016 which showed that a 7-gene APM signature of (HLA-A/B/C, B2M, TAP1, TAP2, and TAPBP), that is contained in the authors 18-gene signature was associated of immune infiltration, TMB, and ORR to immune therapy in kidney. Many of the figures presented by the author duplicate or extend that article.

Some of the figures of APM and APS are redundant with Şenbabaoğlu et al., and I would suggest that these should be moved to the supplement. Instead could the authors should provide a more in-depth discussion of the TIGS scores, IFN etc. Figure 1 should be moved to the supplement.

Are the APS results different to the 7-gene signature described by Şenbabaoğlu et al., There should be a comparison of the performance of the 7-gene Şenbabaoğlu and the 18-gene signature here.

Which cells (CIBERSORT scores) or subsets of the IIS scores are most associated with the APS scores.

The authors should compare their TIGS/APS to recent scores for immune presentation PHBR scores for MHC I and II (Marty et al., 2017 and 2018). https://www.cell.com/cell/pdf/S0092-8674(18)31109-7.pdf https://www.cell.com/cell/pdf/S0092-8674(17)31144-3.pdf

Does APS correlate with PHBR I alone or does it capture both PHBR scores for MHC class I and II? Does TIGS outperform a product of PHBR and TMB?

The authors recently stated that TMB is associated with gender (Wang et al., 2019b). What is the association between TIGS, APS and gender.

In the Introduction, some of the statements are over-generalized and do not reflect the complexity in defining good predictors of immunotherapy response. Whilst a correlation exists, TMB does not always predict response, neither does TIL. Some cancers (e.g. renal) with high immune infiltrate have poor response. Please edit the Introduction to reduce broad over simplifications or generalizations.

Reviewer #3:

Wang et al., describe a new computational method to identify responders for immune checkpoint inhibitors (ICI) using gene expression data from the TCGA database. The authors calculate a tumor immunogenicity score (TIGS) by combining tumor mutational burden (TMB) with antigen processing and presentation machinery (APM) gene signatures. They describe APM signatures across cancer types in TCGA and correlate APM with other gene expression pathways and immune cell infiltration across cancers. They next evaluate the ability of TIGS to predict response to ICI and show improved predictions using this method, compared to TMB alone, the TIDE method by Liu and colleagues, and several biomarkers (PDL1, CD8, etc.). This study is timely given the broad interest in predicting clinical responses to ICI therapy, and the concept of combining antigen presentation gene expression with TMB is also novel and interesting. However, the study is lacking in benchmarking data that support the use of the APM gene signature, and in comprehensive comparisons to prior gene signatures that also synergize with TMB in predicting immune response to ICI. It is also unclear whether the authors have evaluated the performance of prior gene signatures combined with TMB, compared to the TIGS method. Without these comparisons, it is difficult for the reader to truly evaluate the value added by this method, and I suspect will result in lower adoption of the method by the community.

1) A major premise of using antigen presentation gene scores is the presence of mutations in a subset of these genes in non-responder patients (i.e. Zaretsky et al., 2016). Therefore, the authors should determine whether their APM analysis is able to actually capture these defects in tumor samples. In other words, if a patient has a b2m mutation, is this captured by the APM signature and TIGS method? Are these instances the major driver of the value of APM analysis, or are changes in expression levels (without mutation or LOH) also predictive of response?

2) The authors have compared the performance of TIGS to several other prediction tools, however this analysis is not fully described, and I have several questions:

- The comparisons to TIDE are interesting. As I understand it, TIDE only takes into account gene expression, and not TMB. In contrast, TIGS takes into account gene expression (APM) and TMB. The authors should show the data for APM signature alone in several of the figures, for example in Figure 5A-C.

- The authors should also clarify in the main text whether TIDE incorporates TMB into their calculations, and if not, the authors should compare the performance of TIDE +TMB to TIGS.

- Similarly, for PDL1, IFNΓ, and CD8 scores, were these also combined with TMB? Or were they used in isolation to predict response (Figure 5)? The question is: what is the real value added – is it the APM score, or combining gene expression with TMB?

- If TIGS remains a better predictor of response compared to TIDE + TMB, the authors should describe in a main figure the performance comparison in all TCGA cancer types, rather than showing the comparison in 3 (2 that currently perform similarly, and 1 where TIGS outperforms). This information, in a main figure, is critical for the reader to understand the value of this new method across many cancers.

- Since APM genes are turned on by the IFN pathway (as the authors discuss), I would like to see more comprehensive comparisons to IFN pathway signature predictions, beyond only the 6 IFNΓ gene signature score taken from the Ayers et al. manuscript. In particular, I would like to see comparisons to the ISG.RS and IFNΓ.GS signatures described in Benci et al., 2019.

3) The authors correlate APM score with immune infiltration (using IIS and TIGER). Given the prior concerns regarding data normalization in the TIGER method (Newman et al., 2017), I would suggest adding an additional comparison using CIBERSORT. The correlation of APM and immune cell infiltration is independently interesting (without the prediction of response rates), and I think it would be useful to dig into this a bit more – i.e. which cell types correlate most with high APM scores?

---

## [Author Response]

Essential revisions:1) The results of Şenbabaoğlu should be cited more fully and their 18 gene APS compared to the 7 gene Şenbabaoğlu APM geneset. The antigen presenting set is all MHC class I. What is the reasoning behind not including MHC class II genes like DRB1, DRB2, CIITA?

As pointed out by the second reviewer, Şenbabaoğlu et al., 2016 performed antigen presentation gene expression signature analysis, our method for analyzing antigen presentation gene expression signature is similar to Şenbabaoğlu et al., 2016 but with different gene list. And this new information about the citation of Şenbabaoğlu et al., 2016 have been included in the first section of Results in the revised manuscript (subsection “APM score definition and pan-cancer analysis”, last paragraph).

In addition, as suggested, we compared the performance of 7 genes in Şenbabaoğlu et al., 2016 with our 18 genes, results suggested that these two methods show strong association in TCGA pan-cancer level (new Figure 1—figure supplement 1) and similar performance in ICI response prediction (Author response image 1).

MHC I are found on the cell surface of all nucleated cells, and function in displaying peptide fragments of proteins from within the cell to cytotoxic CD8^+^ T cells. MHC II are normally found only on professional antigen-presenting cells such as dendritic cells. The antigens presented by MHC II are derived from extracellular proteins. The anti-cancer immune response against mutated peptides (neoantigen) is primarily attributed to MHC-I-restricted cytotoxic CD8^+^ T cell responses. MHC-II-restricted CD4^+^ T cells also drive anti-tumor responses, however their contribution to neoantigen presentation is not clear.

This manuscript primarily focused on the cytosolic or endogenous neoantigen presentation pathway mediated by MHC I, this does not mean that the potential neoantigen presentation by MHC II is not important. We agree with the reviewers that MHC II presentation could also contribute to the immunogenicity of cancer cells. And the related new discussion has also been included in the revised Discussion part (Discussion, fifth paragraph).

2) TIGS should be compared to TIDE + TMB. Inflamed gene expression signatures have not been shown to be a highly predictive biomarker so I think that comparisons to interferon signatures will show superiority but not guide any advance in our thinking.

TIDE reflect gene expression signatures of T cell dysfunction and exclusion, it may do not have a similar rationale to be combined with TMB as APS in this study. As suggested, we now include the combination of TIDE and TMB in ICI clinical response prediction (Author response image 1). In the Snyder et al., 2017 dataset, combination of TIDE and TMB still show poor ICI response prediction.

3) Multiple reviewers were perplexed by why high TIGs is associated with good outcome in some tumor types but poorer outcomes in other tumor types (Figure 3B). Expand the discussion here.

Most TCGA cancer patients have not been treated with immunotherapy, and the prognosis of these cancer patients are influenced by many factors. Different cancer types could have different prognosis in regards to high APS, TMB or TIGS.

Meta-analysis with all TCGA cancer types suggests that APS is not associated with cancer patients’ prognosis (new Figure 1B), patients with high TMB tends to have poor prognosis (new Figure 2—figure supplement 3), and this observation is similar to previous studies (Owada-Ozaki et al. Prognostic Impact of Tumor Mutation Burden in Patients With Completely Resected Non–Small Cell Lung Cancer: Brief Report. J Thorac Oncol. 2018 Aug;13(8):1217-1221; McNamara et al. Prognostic and predictive impact of high tumor mutation burden (TMB) in solid tumors: A systematic review and meta-analysis. Annals of Oncology 30 (Supplement 5): v25–v54, 2019).

TMB reflect tumor antigenicity, also predict improved survival after immunotherapy. However in cancer patients not treated with immunotherapy, high TMB tends to be associated with poor prognosis, probably because tumor accumulate mutation during progression due to genome instability, and consequently high TMB is usually associated with late stage of cancer.

High TIGS also tends to associated with poor prognosis. This new data and discussion have now been included in the revised manuscript (new Figure 3B). The poor prognosis associated with high TIGS in cancer patients not treated with immunotherapy may be due to similar mechanism as high TMB. This new data analysis and discussion have now been included in the revised manuscript (subsection “Tumor immunogenicity score (TIGS) definition and pan-cancer profiling”, last paragraph).

4) Improve the Cox regression analysis as suggested and make the definition of high or low marker expression consistent throughout.

Cox regression analysis has now been improved as suggested (see revised Figure 1B, Figure 2—figure supplement 3 and Figure 3B). The definition of high and low markers have now been consistent, and Figure 2 and Figure 2—figure supplement 1 has been edited based on new definition.

5) The suggestion to apply the new analyses to many cancer types for ICI response prediction is limited by the available datasets that have all of the required information. The 3 datasets analyzed are what is available and these have been done.

Thanks for this point, we replied the third reviewer as suggested.

6) Indicate the source or reference for their linear correlation formula – "objective response rate = 21.4 ×TIGS – 2.7,".

This is based on the linear regression model of Figure 4, and this information has now been included in the revised manuscript (Discussion, third paragraph).

7) If a patient has a beta2m mutation, is this captured by the APM signature and TIGS method?

Our original method for calculating APM score was based on mRNA expression, it does not directly capture mutation status of APM genes. If a patient has a loss of function beta2m mutation, theoretically, this patient will lose the ability to present antigens through MHC I to immune system, and in the revised APS quantification method, the APS will be reset to zero. Loss of function mutation in beta2m is very rare, and we could not find it in both TCGA and three ICI datasets. Therefore changes in expression level appear to be major driver for APS differences.

Reviewer #1:[…] 1) The definition of high or low marker expression should be consistent throughout. For example, the authors defined patients with APS of first quartile as "APS-High", and those at the fourth quartile as "APS-Low". In contrast, they defined patients with TIGS above the median as "TIGS-high", and the remaining as "TIGS-low". Similar issues exist for the definition of TMB. Please make the correction and explain the rationale behind.

We thank the reviewer for this point. As suggested, we now use the median value as cutoff throughout this study, patients with APS or TMB or TIGS values above the median value were defined as APS-high or TMB-high or TIGS-high respectively. The Figure 2 and Figure 2—figure supplement 1 have thus been corrected based on this new cutoff. Generally the new data do not show apparent difference compared with original one.

2) In Figure 3B, Cox regression analysis show that high TIGS is significantly associated with poor survival of patients in several types of malignancies such as adrenocortical carcinoma (HR=5.23, p=0.00105), Kidney Chromophobe (HR=89.9, p=0.01408), Thymoma (HR=8.22, p=0.00198)…etc. How to explain this phenomena given that TIGS reflects tumor immunogenicity and high TIGS predicts favorable prognosis in patients following immunotherapy.

We thank the reviewer for this point. Most TCGA cancer patients have not been treated with immunotherapy, and the prognosis of these cancer patients are influenced by many factors. Different cancer types could have different prognosis in regards to high APS, TMB or TIGS.

Meta-analysis with all TCGA cancer types suggests that APS is not associated with cancer patients’ prognosis (new Figure 1B), patients with high TMB tends to have poor prognosis (new Figure 2—figure supplement 3), and this observation is similar to previous studies (Owada-Ozaki et al. Prognostic Impact of Tumor Mutation Burden in Patients With Completely Resected Non–Small Cell Lung Cancer: Brief Report. J Thorac Oncol. 2018 Aug;13(8):1217-1221; McNamara et al. Prognostic and predictive impact of high tumor mutation burden (TMB) in solid tumors: A systematic review and meta-analysis. Annals of Oncology 30 (Supplement 5): v25–v54, 2019).

TMB reflect tumor antigenicity, also predict improved survival after immunotherapy. However in cancer patients not treated with immunotherapy, high TMB tends to be associated with poor prognosis, probably because tumor accumulate mutations during progression due to genome instability, and consequently high TMB is usually associated with late stage of cancer.

High TIGS also tends to associated with poor prognosis. This new data and discussion have now been included in the revised manuscript (new Figure 4B). The poor prognosis associated with high TIGS in cancer patients not treated with immunotherapy may be due to similar mechanism as high TMB. This new data analysis and discussion have now been included in the revised manuscript (subsection “Tumor immunogenicity score (TIGS) definition and pan-cancer profiling”, last paragraph).

3) In this study, the authors demonstrate an improved predictive power of TIGS in ICI clinical response when compared to TMB and other gene expression profiling based biomarkers, such as TIDE, IIS, IFNΓ. The authors should discuss the potential mechanisms underlying the superior performance of TIGS for immunotherapy clinical response prediction. The authors state "Furthermore, our linear correlation formula – objective response rate = 21.4 ×TIGS – 2.7, – can be used to.…". Please also indicate the source or reference for this formula.

TIGS captured the two key aspects of tumor immunogenicity, tumor antigenicity and antigen presentation, and theoretically this tumor immunogenicity score should have improved prediction power compared to biomarkers that only reflect tumor antigenicity (such as TMB), or tumor immune environment (such as TIDE IFNΓ). As suggested, this new discussion has now been included (Discussion, first paragraph). In the Results section, we also describe the rationale of TIGS (subsection “Tumor immunogenicity score (TIGS) definition and pan-cancer profiling”). The linear correlation formula was based on data in Figure 4C, and this information has been included in the Discussion part (third paragraph).

Reviewer #2:This well-written article describes a score, the Tumor Immunogenicity Score (TIGS) which combines tumor mutation burden (TMB) and a gene set score of 18 genes associated with Antigen Presentation Machinery (APM) score (APS). The 18 genes were described by Leone et al., 2013) and are PSMB5, PSMB6, PSMB7, PSMB8, PSMB9, PSMB10, TAP1, TAP2, ERAP1, ERAP2, CANX, CALR, PDIA3, TAPBP, B2M, HLA-A, HLA-B, and HLA-C. The author show that the TIGS (product of the natural log of TMB and the normalized APS) marginally outperforms TMB or APS in A) linear associated with Objective Response Rate (ORR), which is a rank order of tissue response to PD1 and PDL1 therapy and B) prediction of response in 2 tissue (Melanoma, Urothelial) in 3 studies (Van Allen, 2015, Hugo et al., 2016 and Snyder et al., 2017).This article represents an extension to Şenbabaoğlu et al., 2016 which showed that a 7-gene APM signature of (HLA-A/B/C, B2M, TAP1, TAP2, and TAPBP), that is contained in the authors 18-gene signature was associated of immune infiltration, TMB, and ORR to immune therapy in kidney. Many of the figures presented by the author duplicate or extend that article.

We agree with the reviewer that Şenbabaoğlu et al., 2016 performed a comprehensive study about cancer immune cell infiltration. Şenbabaoğlu et al., 2016 also reported that elevated 7-APM gene expression signature could predict clinical response to Nivolumab (anti-PD-1) in a ccRCC clinical trial, however, this clinical trial only has 6 patients, and their conclusion need to be validated with more patients. Our study focused the immunogenicity of tumor cells, and propose to combine antigen presentation and tumor mutation burden together in quantifying tumor immunogenicity, and thus the major ideas and conclusions of our study are different from Şenbabaoğlu et al., 2016

Some of the figures of APM and APS are redundant with Şenbabaoğlu et al., 2016 and I would suggest that these should be moved to the supplement. Instead could the authors should provide a more in-depth discussion of the TIGS scores, IFN etc. Figure 1 should be moved to the supplement.

We prefer to keep Figure 1 as main figure, Şenbabaoğlu et al., 2016 did show the comparison of APM between normal vs. cancer in 15 TCGA cancer types in their Supplementary Figure 11. The methods for APM calculation have some similarity, however, the focus and data of our Figure 1 and their Supplementary Figure 11 are different. Their study focused on the comparison of APM between cancer vs. normal, however our study focused on the pan-cancer distribution of APS, and this is critical for further compare the pan-cancer ORR in Figure 4. We include the citation of Şenbabaoğlu et al.’s Supplementary Figure 11 data, which show the comparison of APM between cancer vs. normal, in the Results section (subsection “APM score definition and pan-cancer analysis, last paragraph).

Are the APS results different to the 7-gene signature described by Şenbabaoğlu et al., 2016 There should be a comparison of the performance of the 7-gene Şenbabaoğlu and the 18-gene signature here.

The APS generated with our 18 genes and with 7-gene signature described by Şenbabaoğlu et al., 2016 are highly correlated (Figure 1—figure supplement 1), they also show similar prediction power in immunotherapy response prediction (Author response image 1).

Which cells (CIBERSORT scores) or subsets of the IIS scores are most associated with the APS scores.

We thank the reviewer for this point. As suggested we now included the correlation analysis between APS and cell type status calculated with both CIBERSORT and IIS (new Figure 2—figure supplement 4).

The authors should compare their TIGS/APS to recent scores for immune presentation PHBR scores for MHC I and II (Marty et al., 2017 and 2018). https://www.cell.com/cell/pdf/S0092-8674(18)31109-7.pdf https://www.cell.com/cell/pdf/S0092-8674(17)31144-3.pdfDoes APS correlate with PHBR I alone or does it capture both PHBR scores for MHC class I and II? Does TIGS outperform a product of PHBR and TMB?

As suggested, we included the comparison between APS and PHBR in the revised manuscript (new Figure 1—figure supplement 1).

Our APS captures the expression level information of MHC genes in patient level. Patient Harmonic Best Rank (PHBR) score represents antigen presentation ability for mutations. Both PHBR I and II scores are determined by the MHC genotypes of patients. We obtained patient-by-mutation PHBR-I/II score matrix for TCGA patients from the author of PHBR-I/II paper and summarized the median for each patient, followed by calculation of the correlation between APS and PHBR sores in pan-cancer level. The result shows that APS do not correlate with both PHBR I and PHBR II scores (new Figure 1—figure supplement 1).

PHBR could not capture the antigen presentation differences that exist in different cancer types, since different cancer types should have exactly the same PHBR status because PHBR are MHC genotypes based, thus PHBR cannot be used to explain the ORR differences in cancer types of different tissue origin. On the contrary, our APS and TIGS are gene expression based, and can be used in ORR prediction in different cancer types (Figure 4).

For ICI clinical response prediction in individual cancer patients, raw sequencing information is not available, and this is required for both HLA typing and mutation calling, thus we reviewed the supplementary files of three immunotherapy datasets and found that only two of them (Van Allen, 2015 and Snyder, 2017) have both MHC-I and mutation information available. We then calculated the PHBR I score for each residue and summarized the median for each patient. The result score was used to predict the immunotherapy response. The analysis process is recorded in https://github.com/XSLiuLab/pypresent/tree/master/icb_analysis. The result shows that TIGS outperform PHBR I alone or PHBR I combined with TMB in immunotherapy clinical response prediction (Author response image 1).

The authors recently that TMB is associated with gender (Wang et al., 2019b). What is the association between TIGS, APS and gender.

In our IJC 2019 paper, we did observe a significant gender difference in TMB’s prediction power in lung cancer (349 patient, 171 men 178 women). Due to lack of sufficient number of patients, we could not draw significant gender difference in other cancer types. The three ICI datasets with both genomic DNA mutation and RNA expression data available are melanoma and urothelial cancer. In these datasets, we do not have sufficient number of patients to investigate the gender difference in biomarker’s performance. Related discussion has been included in the revised manuscript (Discussion, fifth paragraph).

In the Introduction, some of the statements are over-generalized and do not reflect the complexity in defining good predictors of immunotherapy response. Whilst a correlation exists, TMB does not always predict response, neither does TIL. Some cancers (e.g. renal) with high immune infiltrate have poor response. Please edit the Introduction to reduce broad over simplifications or generalizations.

We thank the reviewer for this point. We edit the Introduction part as suggested (first paragraph).

Reviewer #3:Wang et al., describe a new computational method to identify responders for immune checkpoint inhibitors (ICI) using gene expression data from the TCGA database. The authors calculate a tumor immunogenicity score (TIGS) by combining tumor mutational burden (TMB) with antigen processing and presentation machinery (APM) gene signatures. They describe APM signatures across cancer types in TCGA and correlate APM with other gene expression pathways and immune cell infiltration across cancers. They next evaluate the ability of TIGS to predict response to ICI and show improved predictions using this method, compared to TMB alone, the TIDE method by Liu and colleagues, and several biomarkers (PDL1, CD8, etc.). This study is timely given the broad interest in predicting clinical responses to ICI therapy, and the concept of combining antigen presentation gene expression with TMB is also novel and interesting. However, the study is lacking in benchmarking data that support the use of the APM gene signature, and in comprehensive comparisons to prior gene signatures that also synergize with TMB in predicting immune response to ICI. It is also unclear whether the authors have evaluated the performance of prior gene signatures combined with TMB, compared to the TIGS method. Without these comparisons, it is difficult for the reader to truly evaluate the value added by this method, and I suspect will result in lower adoption of the method by the community.1) A major premise of using antigen presentation gene scores is the presence of mutations in a subset of these genes in non-responder patients (i.e. Zaretsky et al., 2016). Therefore, the authors should determine whether their APM analysis is able to actually capture these defects in tumor samples. In other words, if a patient has a b2m mutation, is this captured by the APM signature and TIGS method? Are these instances the major driver of the value of APM analysis, or are changes in expression levels (without mutation or LOH) also predictive of response?

We thank the reviewer for this point. Our original method for calculating APM score was based on mRNA expression, it does not directly capture mutation status of APM genes. If a patient has a loss of function b2m mutation, theoretically, this patient will lose the ability to present antigens to immune system, and in the revised APM quantification method, the APS will be reset to zero. Loss of function mutation in b2m is very rare, and we could not find it in both TCGA and three ICI datasets. Therefore changes in expression level appear to be major driver for APM differences.

2) The authors have compared the performance of TIGS to several other prediction tools, however this analysis is not fully described, and I have several questions:- The comparisons to TIDE are interesting. As I understand it, TIDE only takes into account gene expression, and not TMB. In contrast, TIGS takes into account gene expression (APM) and TMB. The authors should show the data for APM signature alone in several of the figures, for example in Figure 5A-C.

We actually showed the performance of APS in ICI response prediction in the original Figure 5D/E/F.

- The authors should also clarify in the main text whether TIDE incorporates TMB into their calculations, and if not, the authors should compare the performance of TIDE +TMB to TIGS.

TIDE reflect gene expression signatures of T cell dysfunction and exclusion, it may be do not have a similar rationale to be combined with TMB as APS in this study. As suggested we combined TIDE with TMB, and this combination still shows poor predictive power in Snyder et al., 2017 dataset (Author response image 1).

- Similarly, for PDL1, IFNΓ, and CD8 scores, were these also combined with TMB? Or were they used in isolation to predict response (Figure 5)? The question is: what is the real value added – is it the APM score, or combining gene expression with TMB?

The combination between APS and TMB has been driven by the rationale that tumor immunogenicity can be divided into two independent steps: neoantigen generation through mutation and antigen presentation through MHC, and this information has been described in the Introduction and Results section. In the original manuscript PDL1, IFNΓ and CD8 score has not been combined with TMB, since we could not find proper rationale for the suggested combination between PDL1, IFNΓ or CD8 with TMB. The key information in Figure 5 is that TIGS measured with combined APS and TMB can outperform known biomarkers in ICI clinical response prediction.

- If TIGS remains a better predictor of response compared to TIDE + TMB, the authors should describe in a main figure the performance comparison in all TCGA cancer types, rather than showing the comparison in 3 (2 that currently perform similarly, and 1 where TIGS outperforms). This information, in a main figure, is critical for the reader to understand the value of this new method across many cancers.

Currently we only have three datasets available for analysis. Since ICI datasets that include patients’ genomic data, gene mRNA expression data and clinical response data are very limited, currently only these three datasets are available.

- Since APM genes are turned on by the IFN pathway (as the authors discuss), I would like to see more comprehensive comparisons to IFN pathway signature predictions, beyond only the 6 IFNΓ gene signature score taken from the Ayers et al. manuscript. In particular, I would like to see comparisons to the ISG.RS and IFNΓ.GS signatures described in Benci et al., 2019.

As suggested, we now included ISG.RS and IFNΓ.GS in ICI clinical response prediction comparison (new Figure 5D/E/F).

3) The authors correlate APM score with immune infiltration (using IIS and TIGER). Given the prior concerns regarding data normalization in the TIGER method (Newman et al., 2017), I would suggest adding an additional comparison using CIBERSORT. The correlation of APM and immune cell infiltration is independently interesting (without the prediction of response rates), and I think it would be useful to dig into this a bit more – i.e. which cell types correlate most with high APM scores?

We thank the reviewer for this point. As suggested, we now include CIBERSORT analysis (new Figure 2—figure supplement 4). The cell types correlated with high APS have now been investigated through both CIBERSORT and IIS analysis (new Figure 2—figure supplement 4).